# Effects of psychosocial support interventions on survival in inpatient and outpatient healthcare settings: A meta-analysis of 106 randomized controlled trials

Timothy B. Smith📧*, Connor Workman📧, Caleb Andrews, Bonnie Barton📧, Matthew Cook, Ryan Layton📧, Alexandra Morrey, Devin Petersen📧, Julianne Holt-Lunstad📧

Department of Psychology, Brigham Young University, Provo, Utah, United States of America

\* TBS@byu.edu

**Data Availability Statement:** All data files are available from the Open Science Framework at https://osf.io/3qydb/.

## Abstract

### Background

Hospitals, clinics, and health organizations have provided psychosocial support interventions for medical patients to supplement curative care. Prior reviews of interventions augmenting psychosocial support in medical settings have reported mixed outcomes. This meta-analysis addresses the questions of how effective are psychosocial support interventions in improving patient survival and which potential moderating features are associated with greater effectiveness.

### Methods and findings

We evaluated randomized controlled trials (RCTs) of psychosocial support interventions in inpatient and outpatient healthcare settings reporting survival data, including studies reporting disease-related or all-cause mortality. Literature searches included studies reported January 1980 through October 2020 accessed from Embase, Medline, Cochrane Library, CINAHL, Alt HealthWatch, PsycINFO, Social Work Abstracts, and Google Scholar databases. At least 2 reviewers screened studies, extracted data, and assessed study quality, with at least 2 independent reviewers also extracting data and assessing study quality. Odds ratio (OR) and hazard ratio (HR) data were analyzed separately using random effects weighted models. Of 42,054 studies searched, 106 RCTs including 40,280 patients met inclusion criteria. Patient average age was 57.2 years, with 52% females and 48% males; 42% had cardiovascular disease (CVD), 36% had cancer, and 22% had other conditions. Across 87 RCTs reporting data for discrete time periods, the average was OR = 1.20 (95% CI = 1.09 to 1.31, $p < 0.001$), indicating a 20% increased likelihood of survival among patients receiving psychosocial support compared to control groups receiving standard medical care. Among those studies, psychosocial interventions explicitly promoting health behaviors yielded improved likelihood of survival, whereas interventions without that primary focus did not. Across 22 RCTs reporting survival time, the average was HR = 1.29 (95% CI

**Funding:** The authors received no specific funding for this work.

**Competing interests:** The authors have declared that no competing interests exist.

**Abbreviations:** CVD, cardiovascular disease; HR, hazard ratio; OR, odds ratio; PRISMA, Preferred Reporting Items for Systematic Reviews and Meta-Analyses; RCT, randomized controlled trial; TAU, treatment as usual.

= 1.12 to 1.49, $p < 0.001$), indicating a 29% increased probability of survival over time among intervention recipients compared to controls. Among those studies, meta-regressions identified 3 moderating variables: control group type, patient disease severity, and risk of research bias. Studies in which control groups received health information/classes in addition to treatment as usual (TAU) averaged weaker effects than those in which control groups received only TAU. Studies with patients having relatively greater disease severity tended to yield smaller gains in survival time relative to control groups. In one of 3 analyses, studies with higher risk of research bias tended to report better outcomes. The main limitation of the data is that interventions very rarely blinded personnel and participants to study arm, such that expectations for improvement were not controlled.

## Conclusions

In this meta-analysis, OR data indicated that psychosocial behavioral support interventions promoting patient motivation/coping to engage in health behaviors improved patient survival, but interventions focusing primarily on patients' social or emotional outcomes did not prolong life. HR data indicated that psychosocial interventions, predominantly focused on social or emotional outcomes, improved survival but yielded similar effects to health information/classes and were less effective among patients with apparently greater disease severity. Risk of research bias remains a plausible threat to data interpretation.

## Author summary

### Why was this study done?

- Medical patients may have difficulty coping with illness. Hospitals, clinics, and health organizations can provide psychosocial support interventions (e.g., calming patients and facilitating treatment adherence) to supplement medical care and possibly improve patient survival.

- Variability exists among psychosocial interventions, and prior evidence about patient survival is mixed; thus, it may be useful to identify factors across research studies that are associated with greater effectiveness.

### What did the researchers do and find?

- A meta-analysis evaluated randomized controlled trials (RCTs) of psychosocial support interventions in medical settings. Separate analyses examined studies reporting patient survival by study end and studies reporting survival rates over time.

- Compared to control groups, those receiving a psychosocial intervention were on average 20% more likely to be alive at study conclusion and had 29% increased likelihood of longer survival, but results varied widely across studies.

- Secondary findings: Study interventions that also included a component supporting health behaviors improved likelihood of patient survival compared with interventions

that did not. Studies with patients having relatively greater disease severity and studies comparing outcomes to groups receiving health information/classes tended to yield nonsignificant gains in survival time. Studies having a low risk of research bias were more likely to report smaller improvements in patient survival.

### What do these findings mean?

- These findings suggest that psychosocial support in medical settings generally promote survival and increase survival time to an extent comparable with rehabilitation programs.

- Intended benefits of psychosocial interventions are to support patients emotionally and to behaviorally cope with their disease.

- Although difficult to accomplish, future research should attempt to keep patients and personnel unaware of group comparisons to reduce the potential for bias due to different expectations for improvement.

## Introduction

Decades ago, researchers found that psychosocial support interventions (e.g., survivor groups and individual nurse support sessions) may improve not only patient quality of life but also patient survival [1,2]. Subsequent evidence regarding patient survival has been mixed [3].

Adequate support among medical patients has been linked to better outcomes, while those that lack adequate support systems have poorer outcomes including greater hospitalization, mortality, and medical costs—such that evaluations of supportive psychosocial interventions have been recommended in healthcare settings [4]. Substantial epidemiological evidence supports the link between psychosocial functioning and health outcomes, including meta-analyses indicating that presence or absence of social support predict all-cause mortality to an extent equivalent to other leading indicators of health (e.g., BMI and smoking cessation) [5–7]. The accumulated research evidence meets the Bradford Hill criteria, establishing low psychosocial support as a causal risk factor for premature mortality [8]. Level of psychosocial functioning has been shown to influence health risk through both emotional coping/resilience and behavioral modeling/motivation [9,10]. However, less is known concerning whether emotional and behavioral support from healthcare professionals can improve medical patients' survival [4]. Given mounting evidence of health consequences of poor psychosocial functioning, the medical community can benefit from evaluating which psychosocial interventions most improve patient survival [11].

Over the past 4 decades, dozens of psychosocial support interventions have been evaluated for medical patients; accumulated literature on the topic is extensive but diverse. These include interventions conducted in patients' homes, in support groups, or via telephone/online conversations. Some psychosocial interventions focus on behavior, explicitly supporting patients' modification of health behaviors. This is based on evidence demonstrating that social support is linked to improved medical adherence [12,13], physical activity [14], sleep [15], and healthcare service utilization [16]. Other psychosocial interventions focus more specifically on

emotion, explicitly supporting patients' coping with distress. Abundant research evidence suggests that psychosocial distress co-occurs with physical disease, with bidirectional relationships that influence disease progression (e.g., appraisal and self-regulation ability) [4]. Research indicates that psychosocial functioning not only affects relevant social capital (e.g., access to health information and improved trust of healthcare) [17] but can also reduce inflammation and improve systemic circulation [18–20]. More specifically, even short-term emotional management interventions can influence inflammatory gene expression [21]. The number of psychosocial interventions with medical patients has multiplied rapidly in recent years, with interventions including multiple overlapping components (e.g., reducing distress and enhancing healthcare utilization). Before the complexity increases further, it would be useful to take stock of extant data by comparing psychosocial interventions across study, intervention, and patient characteristics.

Prior meta-analyses of psychosocial support interventions have evaluated patient survival [22–43]; however, these were susceptible to error due to low numbers of studies included (range = 1 to 36, $M$ = 11.2). Also, few previous meta-analyses have identified effective/ineffective intervention attributes, and most have had limited scope (e.g., breast cancer survivor groups). Although specificity in research is usually optimal, an unintended consequence has been ignoring the reality that professionals across medical specialties use similar psychosocial interventions. Thus, to evaluate differences across contexts, we have conducted what to our knowledge is the largest meta-analytic review to date, including 3 times the number of studies of any prior meta-analysis that we could locate on the topic. We sought to evaluate the overall degree to which psychosocial support interventions improve survival among patients receiving curative or rehabilitative care—and to specifically compare psychosocial interventions emphasizing behavioral support (e.g., modeling/motivation to engage in health behaviors such as physical activity) with those focused primarily on social/emotional support (e.g., emotional resilience following surgery). We also investigated outcome differences across study risk of bias and (a) study characteristics: duration of intervention, length of follow-up, type of control group, and patient psychosocial improvement; (b) intervention type: group meetings, telephone/online support, home visits, and family inclusion; and (c) patient characteristics: age, gender, disease, and mortality rate.

## Methods

### Search strategy

This study is reported as per the Preferred Reporting Items for Systematic Reviews and Meta-Analyses (PRISMA guidelines [44]; S1 Checklist). We sought published and unpublished studies written in any language investigating the effects of psychosocial support interventions on medical patient survival. All authors participated in searching studies completed between January 1980 and October 2020, accessed using Embase, Medline, Cochrane Library, Alt HealthWatch, CINAHL, PsycINFO, Social Work Abstracts, and Google Scholar. To locate all relevant articles, we used an extensive list of search terms, manually examined the reference sections of both prior reviews and studies meeting the inclusion criteria, and contacted authors of included studies (S1 Text).

### Study selection

The meta-analysis included randomized controlled trials (RCTs) reporting data of medical patients' survival as a function of a real-time intervention providing psychological, emotional, and/or social support. We included studies of patients with a health condition likely to result in death if untreated, and who were recruited from healthcare settings (e.g., hospitals,

rehabilitation clinics, or inpatient/outpatient databases). We excluded patients with solely mental health disorders (e.g., anxiety or dementia) because those conditions contribute indirectly to mortality, and we also excluded mortality resulting from accident, suicide, or violence as well as mortality data combined with morbidity/hospitalization.

As the majority of psychosocial support interventions described in the literature involve multiple components, we included interventions with mixed components (e.g., group psychotherapy, nurse visits, and telephone support) and coded for differences to compare outcomes. We excluded those providing only psychoeducation or disease management and those consisting solely of one-on-one psychotherapy, which historically has been a distinct kind of intervention deserving separate systematic review. We similarly excluded hospice or palliative care interventions which deserved separate review because of their focus on improving quality of life, not necessarily length of life, which is the observed outcome of this meta-analysis specific to curative and rehabilitative care. S1 Table provides detailed inclusion/exclusion criteria.

## Data analysis

A team of 2 raters coded each article; subsequently, another team of 2 raters independently coded the same article. Teams resolved discrepancies through manuscript scrutiny until achieving consensus. Coders extracted (a) number of participants with composition by gender and average age; (b) length of intervention and follow-up; (c) type of intervention; and (d) multiple indicators of study risk of bias. Effect size data were hazard ratios (HRs) and odds ratios (ORs); when studies reported other values (e.g., regression coefficients or Cohen's *d*), we transformed them to OR using multiple effect size calculators available online. Data were extracted from the longest follow-up period; when studies contained multiple effect sizes at the same time point (e.g., across subsamples), averaged values were weighted by SE. When reports explicitly tracked mortality but no participants died in either condition, we coded the effect size as OR = 1. We sought effect sizes from multivariable models but calculated OR from survival frequency counts when statistical models were unreported. Stata 16, SPSS 25, and Comprehensive Meta-Analysis 3 were used to calculate random effects weighted models in data aggregation and in subsequent subgroup analysis and meta-regressions.

Our data analysis plan (S2 Text) was to (a) report descriptive statistics of study characteristics; (b) calculate random effects weighted omnibus HR and OR values and also indicators of between-study heterogeneity ($Q$ and $I^2$); (c) conduct subgroup analysis across intervention type (behavior focused versus social/emotional focused); (d) report meta-regressions separately for study, intervention, and patient characteristics; and (e) estimate the likelihood of publication bias. We did not prespecify which variables to include in the meta-regressions but clustered them according to study, intervention, and participant characteristics. We reported a subgroup analysis contrasting behavioral support with social/emotional support as a result of reviewer feedback, not as a prespecified analysis. We prospectively planned to evaluate the likelihood of publication bias estimates using funnel plots, the trim and fill method, and Egger and Peters regression tests. This meta-analysis is registered with Open Science Framework (3nj8u), with data available at https://osf.io/3qydb/.

## Results

### Description of included studies

We located 42,054 studies and screened 909 using the full text (Fig 1). Nonredundant effect sizes were extracted from 106 RCTs [1–3, 45–147] conducted in locations as follows: 50 (47%) in Europe, with 22 in Scandinavia, 11 in the United Kingdom, 6 in the Netherlands, 4 in Germany, and 7 other; 35% in North America, with 28 in the United States and 10 in Canada; 10

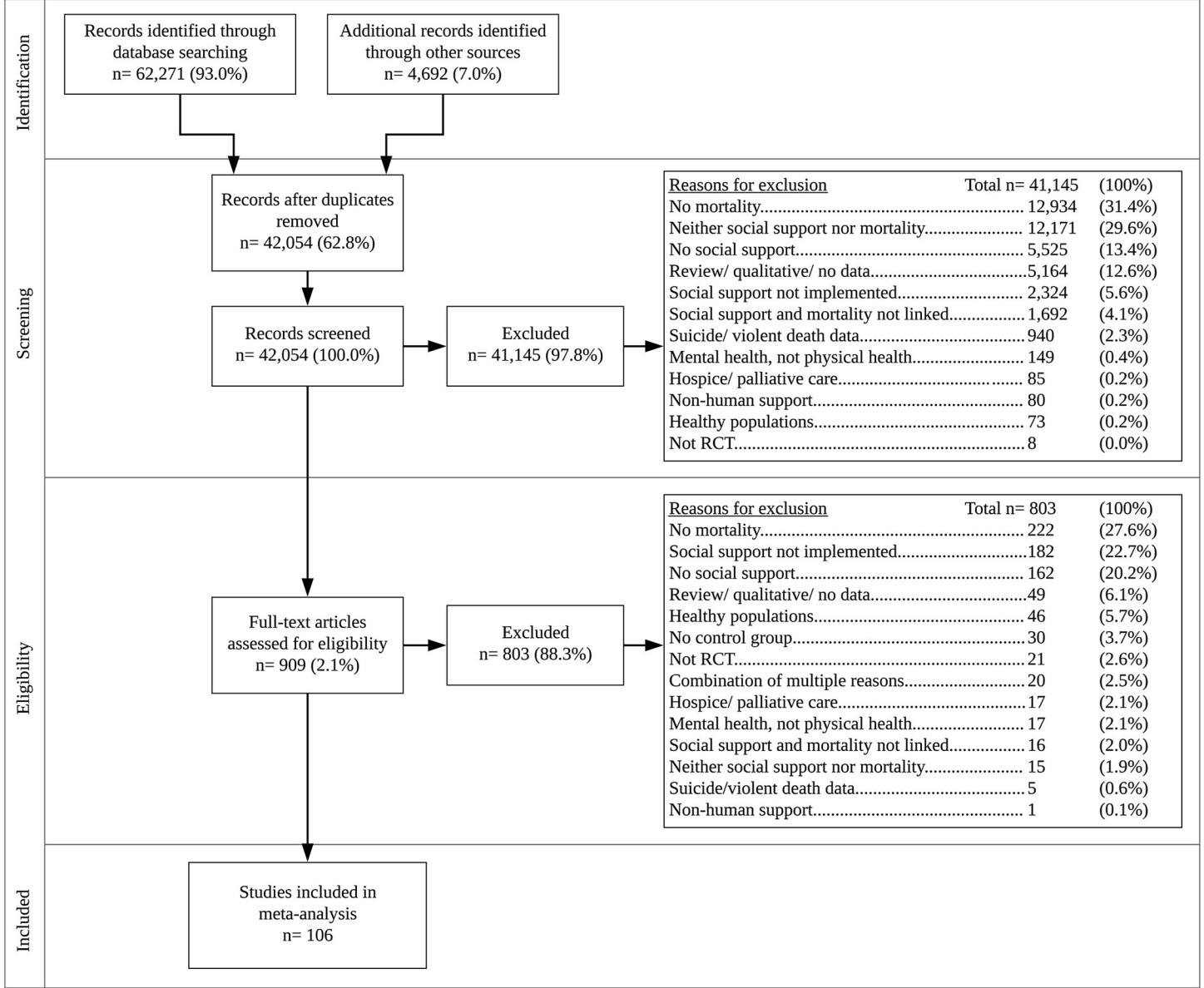

**Fig 1. PRISMA flow diagram of study selection process.** PRISMA, Preferred Reporting Items for Systematic Reviews and Meta-Analyses; RCT, randomized controlled trial.

in Asia; 6 in Australia; and 2 in Africa. Data involved a total of 40,280 participants, whose average age was 57.2 years (SD = 9.9, range = 11 to 78), with an average of 52% females and 48% males. Across all studies, 81 (76%) involved medical outpatients, 20 (19%) recruited hospitalized inpatients, and 5 (5%) involved both. Patients had cardiovascular disease (CVD) in 44 studies (42%), cancer in 38 (36%) studies, or other conditions in 24 (22%) studies; a total of 102 (96%) reported all-cause mortality, with 2 reporting CVD mortality, 1 reporting cancer mortality, and 1 reporting HIV-related mortality.

Regarding intervention focus, 34 of the 106 studies (32%) provided psychosocial behavioral support, explicitly focusing on health behaviors, and 72 (68%) focused on social/emotional support. Many studies included only 1 form of intervention: 46 (43%) in-person group

**Table 1. Study characteristics by type of survival data reported.**

| Variable | OR | | HR | | | |
|---|---|---|---|---|---|---|
| | **Mean** | **SD** | **Mean** | **SD** | *t* | *p* |
| *Participants* | | | | | | |
| Number of participants | 331 | 503 | 571 | 1,276 | 0.9 | 0.40 |
| Participant average age | 57.5 | 10.0 | 55.7 | 9.9 | 0.7 | 0.46 |
| Participant % female | 47.2 | 30.8 | 69.7 | 31.4 | 3.0 | 0.003 |
| Participant % attrition | 9.8 | 11.2 | 9.9 | 15.5 | 0.1 | 0.97 |
| Participant % mortality | 8.2 | 11.3 | 36.7 | 31.2 | 4.0 | 0.001 |
| *Interventions* | | | | | | |
| Number of sessions | 12.2 | 14.3 | 24.3 | 18.2 | 2.8 | 0.002 |
| Minutes of each session | 85.4 | 48.7 | 77.3 | 35.4 | 0.7 | 0.51 |
| Months of intervention | 6.9 | 8.3 | 10.0 | 6.5 | 1.6 | 0.12 |
| Months of follow-up | 15.6 | 23.3 | 65.6 | 58.1 | 3.9 | 0.001 |
| Year initiated | 2003 | 9.0 | 1996 | 8.1 | 3.3 | 0.002 |

Note: Independent samples *t* tests compared 22 studies reporting HR data with 84 studies reporting only OR data; 3 studies reporting both metrics are included in the HR data.

HR, hazard ratio; OR, odds ratio.

meetings, 11 (10%) telephone/online sessions, 9 (8%) home visits, and 7 (7%) in-person individual sessions; the remaining 34 (32%) provided a combination of formats. Interventions were conducted by nurses or medical staff in 37 studies (35%), social workers or mental health professionals in 32 studies (30%), peers with the same medical condition in 4 studies (4%), combinations of those groups in 24 studies (23%), and family members in 1 study, with 8 studies unspecified. Across all interventions, 67 (63%) intended to foster patient social relationships with previously unknown individuals; 20 (19%) provided support only from professional staff; and although 34 (32%) invited family members, only 10 of those focused on family–patient relationships. On average, each intervention session lasted 83 minutes (SD = 46.1; excluding 1 day-long intervention [97]), with 14.7 total sessions (SD = 15.9) over 7.5 months (SD = 8.0). Researchers followed participants after the intervention for an average of 25.6 months (SD = 38.5), during which an average of 13.6% of participants died (SD = 20.3).

Of the 106 RCTs, 87 reported survival data for discrete time periods (transformed to OR), and 22 reported data in terms of survival time (HRs), with 3 studies [74,85,124] reporting both metrics. These types of studies differed in several ways (Table 1). Compared with studies reporting only OR data, studies reporting HR data tended to have an earlier date of initiation and longer total duration, with more female participants, more sessions, longer follow-up, and a correspondingly higher proportion of patient mortality by study conclusion. S2 and S3 Tables contain detailed information about individual studies by data type.

## Main analyses

Across 87 observations at fixed time periods, the average was OR = 1.20 (95% CI = 1.09 to 1.31, $p < 0.001$), indicating a 20% increased likelihood of survival for intervention participants compared to controls. However, the observed effects differed ($Q = 9.3$, $p = 0.002$) between the 31 psychosocial behavioral support interventions having an explicit focus on improving coping/motivation to engage in health behaviors (OR = 1.35, 95% CI = 1.20 to 1.52, $p < 0.001$) and the 56 interventions emphasizing social/emotional support (OR = 1.01, 95% CI = 0.87 to 1.16, $p = 0.94$). The effect sizes for both kinds of interventions varied substantially, with broad confidence intervals (S1 and S2 Figs). However, in separate analyses specific to effect size

heterogeneity, the percentage of variance explained by between-study heterogeneity was estimated to be zero for both the 31 psychosocial interventions based on behavior support ($I^2 = 0.0$; $Q_{(30)} = 27.9$, $p = 0.57$) and the 56 focused on social/emotional support ($I^2 = 0.0$; $Q_{(55)} = 47.0$, $p = 0.77$). Given the absence of between-studies heterogeneity, no further analyses were conducted with OR data.

The 22 RCTs reporting data in terms of survival time averaged HR = 1.29 (95% CI = 1.12 to 1.49, $p < 0.001$), indicating a 29% increased likelihood of longer survival compared to controls (S3 Fig). As only 4 of the 22 studies focused on supporting health behaviors, we did not analyze subgroup differences. Since the HR data were characterized by a moderate percentage of between-study heterogeneity ($I^2 = 54.0$; $Q_{(21)} = 45.7$, $p = 0.001$), we conducted meta-regressions to evaluate possible moderation by study, intervention, and patient attributes.

## Meta-regressions of study, intervention, and patient characteristics

Due to the limited number of studies ($k = 22$), we evaluated study, intervention, and participant characteristics in 3 separate meta-regression models of HR data. The first model, which evaluated study characteristics (Table 2), explained 37.5% of the variance in effect sizes and reached statistical significance ($p = 0.014$). The model included 2 significant predictors: control group type ($\beta = -0.42$, $p = 0.048$) and estimated risk of research bias ($\beta = 0.470$, $p = 0.018$). The 8 studies in which control group members received health information/classes in addition to treatment as usual (TAU) averaged HR = 1.14 (95% CI = 0.92 to 1.40, $p = 0.23$), but the 14 studies with only TAU controls averaged HR = 1.38 (95% CI = 1.17 to 1.62, $p < 0.001$). Studies with relatively higher risk of research bias tended to report improved patient survival as a result of the intervention; given that finding, we included risk of bias in subsequent meta-regression models.

The second meta-regression predicted HR data based on the type of intervention (Table 3). The model explained 10.3% of the variance in effect sizes and did not reach statistical significance ($p = 0.69$). Different kinds of interventions tended to yield similar likelihood of patient survival.

The third meta-regression predicted *HR* data from patient characteristics (Table 4). The model explained 41.0% of the variance in effect sizes ($p = 0.025$). One variable in the model reached statistical significance: Interventions with patients having more advanced disease severity (marked by percentage of patients dying per month) tended to yield lower effect sizes ($\beta = -0.61$, $p = 0.007$). That is, patients with greater disease severity tended to experience reduced benefits from a psychosocial intervention compared to participants in studies with relatively lower disease severity. To put this finding into perspective, 9 studies in which $\geq 0.5\%$ of patients died per month averaged HR = 1.13 (95% CI = 0.95 to 1.34, $p = 0.16$), but 11 studies

**Table 2. Random effects meta-regression of HR estimates of study characteristics on patient survival.**

| Variable | $R^2$ | B | SE | p | $\beta$ |
|---|---|---|---|---|---|
| **Study characteristics** | **0.375** | | | **0.014** | |
| Intervention in months | | −0.004 | 0.010 | 0.69 | −0.083 |
| Follow-up in months | | −0.001 | 0.001 | 0.48 | −0.157 |
| Psychosocial improvement achieved[1] | | −0.087 | 0.084 | 0.30 | −0.234 |
| Control group receiving health information[2] | | −0.270 | 0.136 | 0.048 | −0.421 |
| Risk of bias[3] | | −0.059 | 0.035 | 0.02 | −0.470 |

[1]Statistically significant improvement on psychosocial measures at the end of the intervention compared to controls.

[2]Comparison of studies with control groups receiving only TAU with control groups that received TAU plus information/classes relevant to their health condition.

[3]Sum of indicators of risk of bias.

$\beta$, standardized beta; B, unstandardized beta; HR, hazard ratio; SE, standard error; TAU, treatment as usual. $k = 21$.

**Table 3. Random effects meta-regression of HR estimates of intervention type on patient survival.**

| Variable | $R^2$ | B | SE | p | β |
|---|---|---|---|---|---|
| **Type of intervention** | **0.103** | | | **0.69** | |
| Family support[1] | | −0.003 | 0.064 | 0.97 | −0.008 |
| Group meetings only | | −0.085 | 0.151 | 0.57 | −0.126 |
| Home visit support only | | −0.271 | 0.214 | 0.22 | −0.270 |
| Telephone/online support only | | −0.157 | 0.226 | 0.49 | −0.145 |
| Risk of bias[2] | | −0.004 | 0.037 | 0.91 | −0.022 |

[1]Degree of inclusion of family/partner in the intervention.

[2]Sum of indicators of risk of bias.

β, standardized beta; B, unstandardized beta; HR, hazard ratio; SE, standard error. $k = 22$.

with lower rates of patient mortality averaged HR = 1.64 (95% CI = 1.37 to 1.97, $p < 0.001$). Risk of bias did not reach statistical significance; we conducted collinearity diagnostics and disconfirmed multicollinearity for all 3 meta-regressions.

## Evaluation of risk of bias

Fig 2 summarizes sources of potential bias across all 106 studies (individual studies reported in S4 Fig). In intervention studies of psychosocial support, both personnel and participants know the conditions of the group to which they are assigned. However, it is difficult to limit personnel and/or participant awareness about the other arm of the study in order to diminish unbalanced expectations for improvement. Such blinding of personnel or participants occurred in very few of the 106 studies evaluated (7% blinding participants, 3% blinding personnel, and 2% blinding both). Thus, the results observed in this meta-analysis do not control for plausible expectation differences between treatment and control groups.

Blinding of outcome assessment was unclear in 44% of studies. Although reports of patient death are reasonably reliable, optimally, researchers would confirm patient mortality through independent records. When independent confirmation does not occur, a plausible threat to study validity is that some participants who researchers are "unable to contact" have died. The impact of missing survival data depends on whether participant attrition remains low and balanced across groups. In this meta-analysis, medical patient attrition across all studies averaged 9.9%, with an average difference of 0.6% between the intervention and control groups, so the risk of bias due to attrition was generally low.

Most studies in this meta-analysis explicitly reported the randomization strategy (64%) and allocation concealment (61%). Participants in the intervention and control groups were

**Table 4. Random effects meta-regression of HR estimates of patient characteristics on patient survival.**

| Variable | $R^2$ | B | SE | p | β |
|---|---|---|---|---|---|
| **Patient characteristics** | **0.410** | | | **0.025** | |
| Average patient age at recruitment | | −0.003 | 0.008 | 0.71 | −0.087 |
| Percentage of female patients | | 0.003 | 0.003 | 0.33 | 0.262 |
| CVD patients | | 0.339 | 0.284 | 0.23 | 0.386 |
| Cancer patients | | −0.136 | 0.181 | 0.45 | −0.204 |
| Patient mortality % per month[1] | | −0.272 | 0.100 | 0.007 | −0.606 |
| Risk of bias[2] | | 0.075 | 0.072 | 0.30 | 0.300 |

[1]Number of patient deaths divided by total number of patients divided by total study months.

[2]Sum of indicators of risk of bias.

β, standardized beta; B, unstandardized beta; CVD, cardiovascular disease; HR, hazard ratio; SE, standard error. $k = 19$.

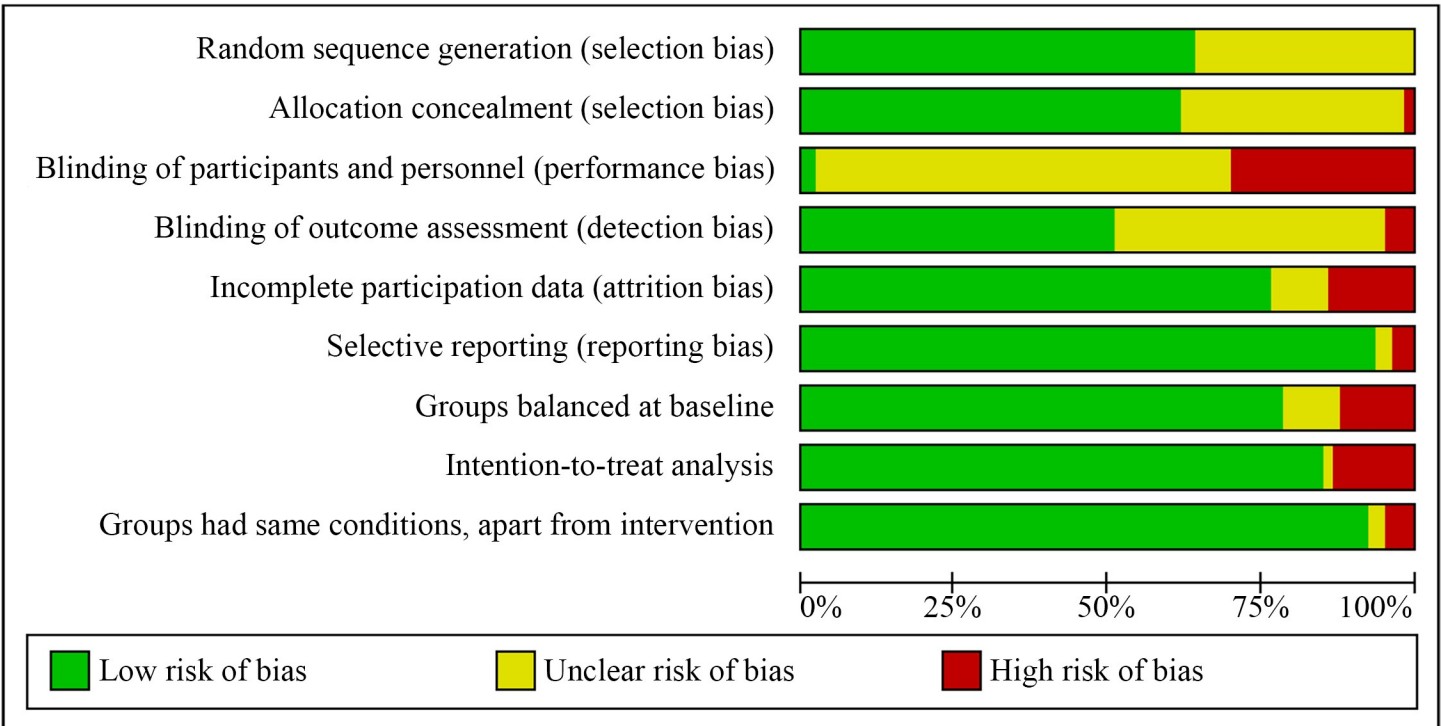

**Fig 2. Risk of bias graph of characteristics across 106 studies.**

typically balanced across variables measured at baseline (78%). The vast majority of studies reported intent-to-treat data (85%) as well as endpoint data on all measures administered (93%).

## Estimate of publication bias

We evaluated the degree to which publication bias may have impacted the overall findings. Begg test, Egger test, and Peters test did not reach statistical significance for either HR or OR data. Inspection of the funnel plots (S5 and S6 Figs) did not suggest more than a few missing studies. Trim and fill analyses [148] of the HR data indicated only 1 missing study using the $L_0$ estimator but 4 missing studies using the $R_0$ estimator. When 4 studies were imputed in the distribution, the results remained statistically significant (HR = 1.22, 95% CI = 1.05 to 1.41, $p$ = 0.009). Trim and fill analysis of the OR data identified no missing studies using the $R_0$ estimator but 8 missing studies using the $L_0$ estimator. When 8 studies were imputed in the distribution, the results of the OR data remained statistically significant (OR = 1.15, 95% CI = 1.03 to 1.29, $p$ = 0.015). Overall, the results of this meta-analysis did not appear to be adversely impacted by publication bias.

## Discussion

### Statement of principal findings

This meta-analysis, including 106 RCTs and 40,280 participants, examined the extent to which different types of psychosocial support interventions increased survival among medical patients receiving curative or rehabilitative care. Overall, the interventions increased odds of survival (OR = 1.20) and relative length of survival (HR = 1.29), with the magnitude of these data being comparable with other tertiary prevention interventions (Fig 3).

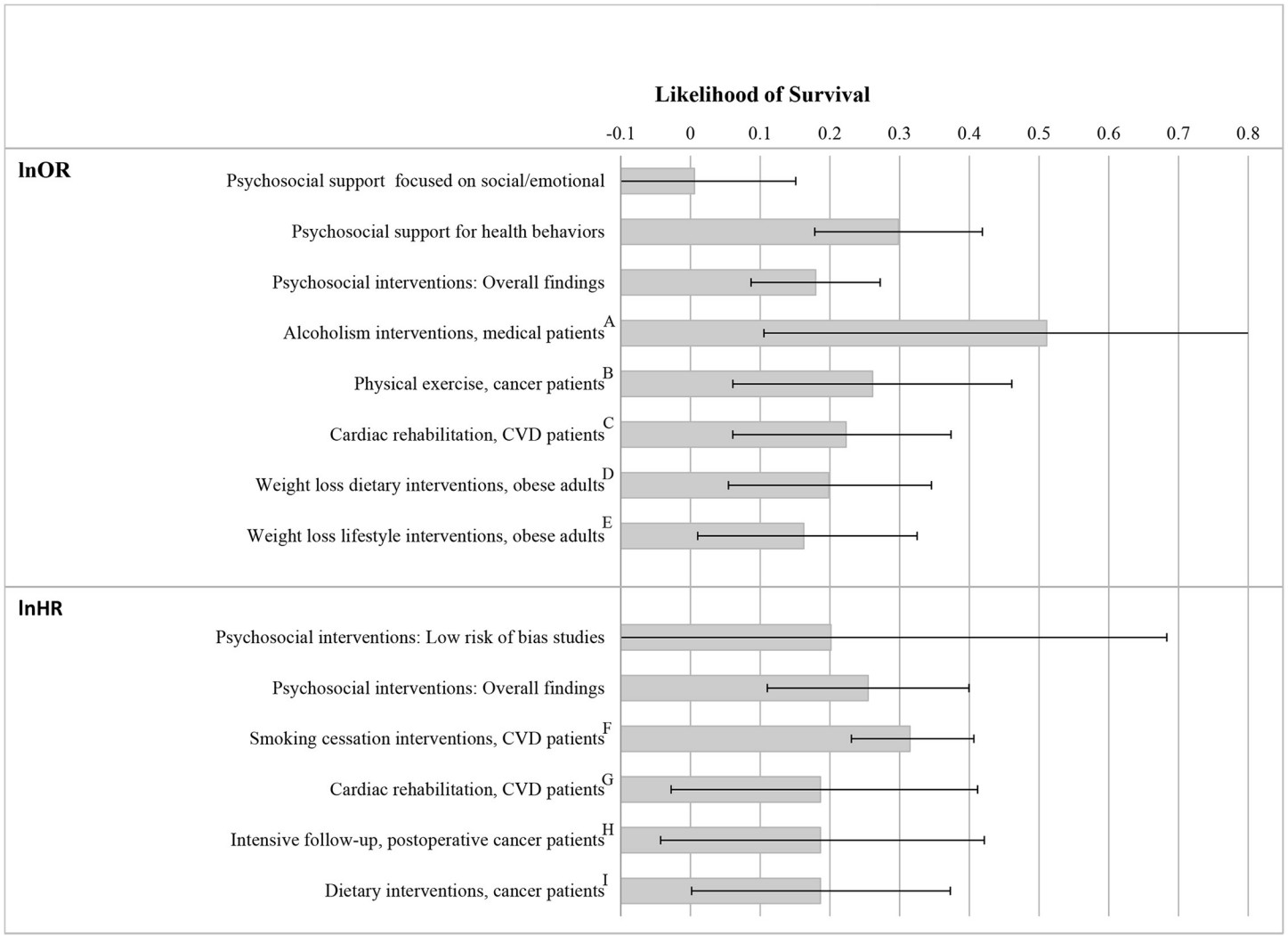

**Fig 3. Comparison of odds (lnOR) and hazards (lnHR) of mortality across several tertiary prevention interventions.** Note: lnOR = natural logarithm of the OR of patient survival. lnHR = natural logarithm of the HR of patient survival. Effect size of 0 indicates no effect, and values above 1 favor the intervention group relative to the control group. Comparison effect sizes and 95% confidence intervals were reported in meta-analyses: A = McQueen et al. [149]; B = Wu et al. [150]; C = Taylor et al. [151]; D = Ma et al. [152]; E = Kritchevsky et al. [153]; F = Mons et al. [154]; G = Taylor et al. [155]; H = Calman et al. [156]; I = Hauner et al. [157]. CVD, cardiovascular disease; HR, hazard ratio; OR, odds ratio.

Across the 87 studies reporting survival data at a fixed point in time, the 31 psychosocial behavioral support interventions (e.g., motivation for treatment adherence) improved the likelihood of patient survival, but the 56 interventions emphasizing social/emotional support yielded results no better than those of control groups. It is unclear whether behaviorally focused interventions are more effective or whether these types of interventions merely involve more components (behavioral and social/emotional), thereby providing greater diversity of support. As only 4 of the studies reporting HR data were explicitly focused on promoting patient health behaviors, a similar subgroup comparison was not advisable until additional studies reporting HR data accrue in the literature. The HR data predominantly represented interventions focused on social/emotional outcomes (18 of 22 studies).

Other differences between the studies reporting OR and HR data can inform data interpretation. A primary difference involves the nature of ORs and HRs. ORs provide a snapshot at a fixed point in time, but HRs reflect changes across time. Moreover, Cox proportional hazards

regression models typically include covariates, such that other variables (e.g., initial health status and socioeconomic status) are less likely to influence the reported outcomes. In terms of our data specifically, the 22 studies reporting HR data tended to have more female participants, twice as many sessions, and 5 more years of patient follow-up, with correspondingly lower rates of patient survival than the 87 studies reporting OR data. Future research is needed to confirm whether interventions with more sessions and longer follow-up yield greater benefits, as recommended in a National Academy of Science report [4].

Analyses with HR data indicated that patient disease severity (percentage of deaths per month) moderated the overall findings. Specifically, studies in which a relatively larger percentage of patients died each month tended to report fewer benefits from the psychosocial intervention in terms of patient survival compared to control conditions. Future research can investigate if the higher mortality rates are a function of more reliable outcomes when death is not uncommon in the distribution. Alternatively, psychosocial interventions might be more effective in improving survival among patients when conducted earlier in the disease trajectory, consistent with effectiveness of other medical treatment.

Meta-regression analyses with HR data indicated that effect sizes did not differ across the format of the intervention (support groups, telephone/online conversations, family involvement, or home visits). However, in one of the meta-regressions, the findings differed as a function of study risk of bias, with studies reporting more robust results also tending to have more indicators of research bias. Having disconfirmed multicollinearity, we cannot account for why that variable reached statistical significance in only one of 3 analyses, but the result provides a caution that qualifies the overall findings reported in the literature. The overall strength of evidence was mixed (Fig 2), with the primary limitation being the neglect of blinding personnel and participants to study conditions. Thus, it is difficult to distinguish between intervention effects and expectation effects when personnel and participants have knowledge of both study conditions. This concern was reinforced by the finding that 8 psychosocial support interventions reporting HR data did not show statistically significant differences from control groups receiving health information/classes.

## Limitations of the study

This meta-analysis has several limitations. First, the results varied widely across individual studies (see S1–S3 Figs). The omnibus results should be interpreted using their confidence intervals. Across the 87 studies reporting OR data, the confidence intervals for individual studies were so wide that there were no nonoverlapping values ($I^2 = 0.0$). The wide confidence intervals for most of these OR studies corresponded with a numerically low percentage of patients who died across studies (8.1%, see Table 1); low mortality rates yielded high standard error values. Second, variability existed in the approach and delivery of support provided in the studies. Psychosocial support was offered via peer support groups, telephone calls, one-on-one nurse sessions, etc., with our statistical contrast being the mixed interventions. Third, only 10 of the 106 RCTs included support from naturally occurring relationships in at least half of the intervention, with 6 of those focusing specifically on family/partners, yet preexisting relationships constituted the epidemiological evidence that precipitated such interventions [5]. Strengthening preexisting close relationships may produce longer-lasting effects due to the chronic and often intimate nature of such relationships [158]; nonetheless, not all patients have supportive social networks. Fourth, we did not evaluate preexisting levels of patient psychosocial support because the literature inconsistently reported such data. Patients with strong social networks tend to fare better than others on multiple clinical markers [20,159] and outcomes [159–161]. Failure to account for preexisting differences in social resources can be corrected in future research [162]. Fifth, although many of the studies reporting HR data included

other variables in statistical models, such as patient age and health status, only 3 of the studies reporting OR data statistically controlled for other variables. The HR estimates were therefore more trustworthy than the OR estimates [43].

## Implications for clinicians, researchers, and policy makers

Prior meta-analyses have reported mixed results [25,27,31,36], some concluding that psychosocial interventions did not improve patient survival [23,33,37]. Therefore, a major contribution of this meta-analysis was to clarify that although the vast majority of studies did not reach statistical significance (96 of 106 [91%]; see S5 and S6 Figs), psychosocial interventions overall tended to benefit survival with results comparable to rehabilitation programs (Fig 3). However, the extent of variability in results across studies suggests that care must be taken during design and implementation to maximize patient outcomes. In particular, this meta-analysis confirmed that the minority of interventions (32%) that explicitly promoted patient motivation/coping to engage in health behaviors tended to improve patient survival, with an observed effect (OR = 1.35) corresponding with a number needed to treat of 19.6. Rather than focus solely on emotional and psychological support, future psychosocial support interventions with medical patients should also address health behaviors (e.g., motivation for treatment adherence). The accumulated data now make it questionable to neglect including behavioral support when planning psychosocial interventions with medical patients receiving curative care.

Given the concerns raised in this meta-analysis about study risk of bias adversely impacting the reported results, future research should specifically address that issue. Although blinding personnel and participants to the other study arm may be challenging, this gap needs to be addressed to advance the science beyond its current state. Other scholars have recommended that future research identify patient existing psychosocial supports and needs, evaluate specific causal pathways influencing disease progression [9,10,163], focus on strengthening naturally occurring relationships [158], and refine interventions utilizing the Multiphase Optimization Strategy [164,165].

Despite the multiple qualifications and concerns raised in this meta-analysis, psychosocial support interventions improved medical patient survival to a degree comparable with other tertiary prevention methods (Fig 3), with the findings being equivalent to a meta-analysis of epidemiological data on the effects of social isolation on mortality [6]. Taken together with prior research documenting that social isolation increases healthcare costs [166] and excessive utilization [16,167], and with increasing social isolation in recent years [168], this meta-analysis urges increased methodological rigor but tentatively supports recommendations [4] to consider psychosocial interventions in promoting health behavior in a public health framework.

## Supporting information

**S1 Checklist. PRISMA 2009 checklist.** PRISMA, Preferred Reporting Items for Systematic Reviews and Meta-Analyses.
(PDF)

**S1 Alternative Language Abstract. Spanish translation of the abstract by Laura Melgarejo Perez and Juan Valladares.**
(PDF)

**S2 Alternative Language Abstract. Traditional Chinese Characters translation of the abstract by Cheng Wai Man.**
(PDF)

**S3 Alternative Language Abstract. Simplified Chinese Characters translation of the abstract by Li Zhen.**
(PDF)

**S4 Alternative Language Abstract. Tamil translation of the abstract by Babu Manuel Abel.**
(PDF)

**S5 Alternative Language Abstract. Italian translation of the abstract by Claudia Mencarelli and Tommaso Cardullo.**
(PDF)

**S6 Alternative Language Abstract. Turkish translation of the abstract by Murat Çakır.**
(PDF)

**S7 Alternative Language Abstract. German translation of the abstract by Samira Herber.**
(PDF)

**S8 Alternative Language Abstract. Indonesian translation of the abstract by Pungki Lupiyaningdyah.**
(PDF)

**S9 Alternative Language Abstract. Arabic translation of the abstract by Sara Abu Al-Samen.**
(PDF)

**S10 Alternative Language Abstract. Portuguese translation of the abstract by Solange Andrezzo and Larissa Vecchi.**
(PDF)

**S1 Text. Literature search strategies and selection criteria.**
(PDF)

**S2 Text. Data abstraction and analyses.**
(PDF)

**S1 Table. PICOT inclusion criteria.**
(PDF)

**S2 Table. Characteristics of 87 psychosocial intervention studies reporting ORs of medical patient survival.** OR, odds ratio.
(PDF)

**S3 Table. Characteristics of 22 psychosocial intervention studies reporting HRs of medical patient survival.** HR, hazard ratio.
(PDF)

**S1 Fig. Forest plot of 56 social/emotional support focused RCTs reporting OR.** OR, odds ratio; RCT, randomized controlled trial.
(PDF)

**S2 Fig. Forest plot of 31 behavioral support RCTs reporting ORs.** OR, odds ratio; RCT, randomized controlled trial.
(PDF)

**S3 Fig. Forest plot of 22 RCTs reporting HRs.** HR, hazard ratio; RCT, randomized controlled trial.
(PDF)

**S4 Fig. Risk of bias summary.**
(PDF)

**S5 Fig. Contour-enhanced funnel plot of 89 RCTs, OR data.** OR, odds ratio; RCT, randomized controlled trial.
(PDF)

**S6 Fig. Contour-enhanced funnel plot of 22 RCTs, HR data.** HR, hazard ratio; RCT, randomized controlled trial.
(PDF)

## Acknowledgments

We thank our research assistants for their many contributions to this project.

## Author Contributions

**Conceptualization:** Timothy B. Smith, Julianne Holt-Lunstad.

**Data curation:** Timothy B. Smith, Connor Workman, Caleb Andrews, Bonnie Barton, Matthew Cook, Ryan Layton, Alexandra Morrey, Devin Petersen.

**Formal analysis:** Timothy B. Smith.

**Funding acquisition:** Timothy B. Smith.

**Investigation:** Timothy B. Smith, Connor Workman, Caleb Andrews, Bonnie Barton, Matthew Cook, Ryan Layton, Alexandra Morrey, Devin Petersen, Julianne Holt-Lunstad.

**Methodology:** Timothy B. Smith.

**Project administration:** Timothy B. Smith, Julianne Holt-Lunstad.

**Resources:** Timothy B. Smith, Julianne Holt-Lunstad.

**Software:** Timothy B. Smith, Connor Workman.

**Supervision:** Timothy B. Smith, Caleb Andrews, Matthew Cook, Alexandra Morrey, Devin Petersen, Julianne Holt-Lunstad.

**Validation:** Timothy B. Smith, Connor Workman, Caleb Andrews, Bonnie Barton, Ryan Layton, Alexandra Morrey, Devin Petersen, Julianne Holt-Lunstad.

**Visualization:** Timothy B. Smith, Connor Workman.

**Writing – original draft:** Timothy B. Smith, Connor Workman, Bonnie Barton, Ryan Layton, Devin Petersen, Julianne Holt-Lunstad.

**Writing – review & editing:** Timothy B. Smith, Connor Workman, Caleb Andrews, Bonnie Barton, Matthew Cook, Ryan Layton, Alexandra Morrey, Devin Petersen, Julianne Holt-Lunstad.

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
