## [Editor Report · Decision Letter 0]

20 Jul 2020

Dear Dr Smith, 

Thank you for submitting your manuscript entitled "Psychosocial Support Interventions and Medical Patient Survival: A Meta-Analysis of 140 Randomised Controlled Trials" for consideration by PLOS Medicine.

Your manuscript has now been evaluated by the PLOS Medicine editorial staff and I am writing to let you know that we would like to send your submission out for external peer review.

Kind regards,

Caitlin Moyer, Ph.D.,

Associate Editor

PLOS Medicine

---

## [Decision Letter · Decision Letter 1]

1 Oct 2020

Dear Dr. Smith,

Thank you very much for submitting your manuscript "Psychosocial Support Interventions and Medical Patient Survival: A Meta-Analysis of 140 Randomised Controlled Trials" (PMEDICINE-D-20-03297R1) for consideration at PLOS Medicine. 

Your paper was evaluated by a senior editor and discussed among all the editors here. It was also discussed with an academic editor with relevant expertise, and sent to three independent reviewers, including a statistical reviewer. The reviews are appended at the bottom of this email and any accompanying reviewer attachments can be seen via the link below:

[LINK]

In light of these reviews, I am afraid that we will not be able to accept the manuscript for publication in the journal in its current form, but we would like to consider a revised version that addresses the reviewers' and editors' comments. Obviously we cannot make any decision about publication until we have seen the revised manuscript and your response, and we plan to seek re-review by one or more of the reviewers. 

We expect to receive your revised manuscript by Oct 22 2020 11:59PM. Please email us (plosmedicine@plos.org) if you have any questions or concerns.

We look forward to receiving your revised manuscript. 

Sincerely,

Caitlin Moyer, Ph.D.

Associate Editor 

PLOS Medicine

plosmedicine.org

1.Prospective analysis plan: Did your study have a prospective protocol or analysis plan? If so, please provide the analysis plan as a supporting information file. Please state whether analyses were pre-planned (either way) early in the Methods section.

2.Data availability statement: “All data files will be available from the BYU Scholar's Archive database at the time of publication (URL to be determined).” Thank you for agreeing to make your data available. If the data are freely or publicly available, note this and state the location of the data: within the paper, in Supporting Information files, or in a public repository (include the DOI or accession number). If the data are owned by a third party but freely available upon request, please note this and state the owner of the data set and contact information for data requests (web or email address). Note that a study author cannot be the contact person for the data.

3.Abstract: Background: The final sentence should clearly state the study question.

4.Abstract: Methods and Findings: Please briefly summarize the patient demographics of those included studies.

5.Abstract: Methods and Findings: Please provide p values for these results, also please provide more description of the results (for example, what are the OR/HR representing here: “Across 112 RCTs reporting data for discrete time periods, the average was OR = 1.15 (95% CI = 1.03 to 1.29; I2 = 23.6, I2 CI 4 to 40). Across 29 RCTs reporting survival time, the average was HR = 1.32 (95% CI = 1.15 to 1.51; I2 = 53.6, I2 CI 29 to70).”)

6.Abstract: Methods and Findings: In the last sentence of the Abstract Methods and Findings section, please describe the main limitation(s) of the study's methodology.

7.Abstract: Conclusions: Please address the study implications without overreaching what can be concluded from the data; the phrase "In this study, we observed ..." may be useful. It is not clear what is meant by “interventions minimally improved patient survival…” in the first sentence- please clarify this to either state that the interventions did improve, or did not improve survival.

Please interpret the study based on the results presented in the abstract, emphasizing what is new without overstating your conclusions. “Psychosocial support programmes can be improved by better meeting patient needs.” this sentence does not seem to be supported by the study’s data or objectives.

9.Author Summary: At this stage, we ask that you include a short, non-technical Author Summary of your research to make findings accessible to a wide audience that includes both scientists and non-scientists. The Author Summary should immediately follow the Abstract in your revised manuscript. This text is subject to editorial change and should be distinct from the scientific abstract. Please see our author guidelines for more information: https://journals.plos.org/plosmedicine/s/revising-your-manuscript#loc-author-summary

10.References throughout: For in-text citations, please place the reference number in brackets before the punctuation, like this [1].

11.Introduction: Line 77-78: “Thus, in conducting the largest meta analytic review to date…” Please qualify this with “to the best of our knowledge” or similar.

12.Methods: Line 88: Please revise this to “This study is reported as per the Preferred Reporting Items for Systematic Reviews and Meta-Analyses (PRISMA) guideline.” Please provide the completed PRISMA checklist as a supporting information file. When completing the checklist, please use section and paragraph numbers, rather than page numbers. 

13.Methods Line 91-92: Please update your search to the present time.

14.Results: Lines 146-148: Please provide the p values associated with “Across 112 estimates using fixed time periods, the average was OR = 1.15 (95% CI = 1.03 to 1.29), indicating a small increased likelihood of survival for intervention participants compared to controls.

15.Results: Lines 156-157: Please provide the OR/HR data with 95% CIs and p values for the findings excluding terminally ill patients or deaths shortly after baseline, between inpatinets and outpatients, by health condition, and attrition vs. mortality, or treatment as usual with our without additional materials. Please refer to the table where these are presented.

16.Results: Lines 167-168: Please also present the HR and 95% CI breakdown for this result, that intervention length was not associated with survival time.

17.Results: LInes 171-174: Please provide the p values for the sensitivity analyses of the high quality studies.

18.Results: Liines 177-181: Please provide p values for psychosocial functioning results.

19. Discussion: Line 316: The reference to the “current pandemic” should be removed.

20.Figure 4: Please define the abbreviations for OR, HR, CVD in the figure legend.

21.Competing interests: Can you please clarify the existing competing interests of the authors? The sentence “no financial relationships with any organisations that might have an interest in the submitted work in the previous three years” is not clear.

22.Supporting information: Please provide titles and legends for each individual table and figure in the Supporting Information.

Comments from the Academic Editor:

1. The Introduction seems to conflate several different constructs (eg., social isolation, loneliness, and social support). And those are different from other constructs in the literature. eg., 

Social integration: 

Seeman TE. Ann Epidemiol 1996;6:442-451. 

Tsai AC, Lucas M, Sania A, Kim D, Kawachi I. Ann Intern Med. 2014 Jul 15;161(2):85-95. 

Tsai AC, Lucas M, Kawachi I. JAMA Psychiatry. 2015 Oct;72(10):987-993. 

Social participation: 

Obembe AO, Eng JJ. Neurorehab Neural Repair 2016;30:384-392.

Kuiper JS, et al. Ageing Res Rev 2015;22:39-57.

As a result, it is unclear to me exactly what kinds of interventions are the focus of this study. The title and Methods refer to "psychosocial support" interventions, but when I look at the list of included studies, I see several CBT interventions (eg., Choi et al ref #63)-- but I would not characterize CBT as a "social support intervention". When I get to the Limitations and find out (line 280) that only 8/140 RCTs focused on naturally-occurring relationships, then I am even less certain as to what kinds of interventions were studied. ENRICHD, for example, was focused on getting study participants to strengthen their existing relationships, and is generally regarded as a prototypical "social support intervention". Does this mean that 132/140 RCTs were more of the "prescribe a friend" type? Lines 134-139 indicate that most interventions were more of the "clinician-delivered support" type.

2. The search needs to be updated. My understanding is that PLOS generally requests for evidence searches to be out of date by no more than 6 months. Should this manuscript make it through peer review, by the time it is published, more than 3 years will have elapsed since the evidence search was conducted.

3. Line 98: The authors restricted analysis to studies of "medical patients". Perhaps they could be more specific about what this term means. Are they referring to inpatients? Outpatients? Or did the included studies need to include study participants with a defined medical diagnosis and/or acute medical illness? (Line 104 ["mental illness as the sole health condition"] seems to imply that included studies were focused on some health conditions.) Any study conducted with a participant sample recruited in a health care setting? If so, does this mean they excluded group interventions occurring outside of health care settings (eg., Alcoholics Anonymous)? I see "self-help group" in the search strategy but not "AA", "Alcoholics Anonymous", or "12 step" (and those terms would normally be included in a systematic review of AA-style interventions, cf. Kelly JF, Magill M, Stout RL. Addict Res Theory 2009;3(17):236-259). They seem to have included "home visits", so I would expect that AA-style interventions were also included--but the search strategy does not seem like it was designed to capture these.

4. Lines 101-104: There are a few things that make the exclusions a bit "squishy". (a) The authors state that they are focused on "disease" but then exclude suicide, mental illness, dementia-- is the implication that mental illness is "not disease"? (b) If the authors excluded studies of mental illness as the "sole health condition", does that mean they included studies that were focused on depression among study participants with another medical condition? I see Berkman/ENRICHD in the list of included studies, so I assume this is true. Please clarify. (c) I am in agreement with R2 that the exclusion of palliative care interventions "not intended to prolong life" seems problematic and should be clarified or amended.

5. Line 111: Does this mean that none of the studies reported data as (for example) standardized effect sizes, regression coefficients, etc.? I find this very, very hard to believe. In any case, if this is true, please indicate here in the text.

6. Line 115: What does "questionable effect size data" mean? Please be more specific here in the text. Examples (eg., in the Appendix) may help.

7. Line 116 and elsewhere: The authors use the term "multivariate" but I believe they mean "multivariable". (cf. Peters TJ. Paediatr Perinat Epidemiol 2008;22(6):506).

8. Line 116: The authors included randomized controlled trials. Why would these studies need to "control for known confounds"? Are the authors referring to randomized studies in which covariate balance was not achieved through randomization? Please clarify here in the text.

9. Line 129 seems to suggest that studies were restricted to either medical inpatients or medical outpatients. (cf comment #3 above)

10. Abstract, Line 147, Line 236, Figure 4, etc.: why are the findings (pooled OR=1.15, pooled HR=1.32) described as "minimally" improving patient survival (eg., "small increased likelihood of survival", "increased odds to a small degree")? I am _not_ suggesting that the authors revise this adjective (eg., "substantially"). I am simply requesting that the authors justify their use of this adjective. I see the comparisons to exercise, cardiac rehab, etc. But I do not find those examples too compelling. Aspirin reduces the risk of CVD events by 15% in relative terms (ie, meta-analysis OR=0.85), but I do not believe that clinicians would characterize that as a minimal improvement. Perhaps in the literature on social support interventions the bar is different?

11. Line 155: All of these subgroup analyses, and the meta-regression, should be described in the Methods section. 

12. Line 158 has a typo ("statistically significant differ")

13. Line 169: I see the N's for the low-quality studies. What were the N's for the high quality studies, and what were the pooled estimates?

I did not see a Risk of Bias assessment table. This would be particularly important to know which studies (OR vs HR data) were in the "high risk of bias" vs "low risk of bias" groups, because the higher quality studies did not reach statistical significance. 

How was risk of bias assessed? This is only mentioned in passing on line 109. eg Did a study get 1 point each for blinding, exclusions, attrition (what threshold?), and then "low risk of bias" = 3 points?

14. Line 175 and Figures 2 & 3: what do the authors mean by "statistically significant improvements on psychosocial variables"? Presumably variables like depression symptom severity, social role functioning, etc. were included. What kinds of variables were excluded? What do the authors mean by "inconsistent/ineffective" in Figure 2? (What if a study was "consistent" but "ineffective", or "inconsistent" but "effective"?) I see that these data are partially represented graphically in Figures 2 & 3, but it would be helpful to have an Appendix table that is analogous to Table 2 showing each study line by line (stratified by "inconsistent/ineffective" vs. "effective"), the "psychosocial variable" assessed (eg., 15-item Hopkins Symptom Checklist), and the efficacy estimate and standard error on the psychosocial variable.

15. Line 184: The average age and percentage of female patients are not study-level characteristics and should not be included in the meta-regression model.

16. Line 284: "many studies did not measure preexisting levels of support"-- I did not see this anywhere in the Results, please add.

Comments from the reviewers:

Reviewer #1: I confine my remarks to statistical aspects of this paper. The general approach is fine, but I have some concerns and suggestions before I can recommend publication.

First - As I was reading along, I thought "why aren't they doing meta-regression?" Unless I missed something, this was first mentioned on p 8. It ought to be mentioned in methods on p. 4 or 5. (The authors say 'subsequent analysis' but that is very vague). 

More specific comments

Lines 50 and 52: It isn't clear what these OR and HR are, exactly. It seems like it is "support of any kind" vs. "no support" but this should be made clear. 

Line 101 What does the sentence starting "Differences were coded ..." mean? What differences? Coded how? 

Line 149 ff I don't think statistical significance of Isquared is particularly useful. Maybe delete the p values and CIs.

Subgroup analysis - I'm unclear on how subgroup analysis was used to conduct hypothesis tests and yield p values and so on. This is what meta-regression doe. But if you do an analysis only on one subgroup, how do you do a tst of it vs. another subgroup? This doesn't make the subgroup analysis wrong, but .... how did the authors get these results?

Fig. 2 is way too small a font. Maybe spread it over several pages. 

Fig 3 I like that there is relatively little text here. But it would be good to add a vertical line at OR = 1 That would make it a lot clearer that these effects are in one direction.

Peter Flom

Reviewer #2: Review PMEDICINE-D-20-03297R1

Psychosocial Support Interventions and Medical Patient Survival: A Meta-Analysis of 140 Randomised Controlled Trials

I welcome the goal to perform a comprehensive meta-analysis to establish the state of knowledge on this topic. The heterogeneity of interventions and medical conditions included however is linked to significant problems in the present work which raise the risk of misinterpretation due to oversimplification. My points include 1) oversimplified presentation of the theoretical background, 2) need for refinements in some of the central extracted data from studies and 3) need to discuss conclusions in a much more modest way.

1) In the introduction the authors aim to establish a link from patient loneliness and lack of social connections to the eventual meta-analysis of a wide range of psychosocial support interventions, including for example several-session telephone disease management programs. The authors need to better explain how and why such interventions would be able to reduce loneliness/increase social connectedness as the assumed mechanism of prolonging survival. It is hard to imagine how low-dosis interventions of this structure that have been included in the review could change an individuals' social relationships so enduringly that survival is significantly prolonged. Yet it seems reasonable to assume a different mechanism: in-person disease management or telephone support with home visits can improve health behavior (therapy adherence, smoking, diet, alcohol consumption etc.), which in turn improves survival. 

2) Inclusion criteria: The authors state that interventions "not intended to prolong life" were excluded. This is a difficult statement because, at least for the majority of the included cancer studies, the study was not conducted with a primary intention of prolonging survival but to enhance quality of life. However, survival was co-assessed as one of many potential outcomes. This needs to be clearly stated. It is difficult to understand why similar studies in palliative care populations were excluded. For example, several early-palliative-care intervention trials reported survival data along a range of outcomes and they are not represented in this review.

Page 6, line 139: The average length of intervention sessions seems very long. Do the authors mean the total length of all sessions?

Abstract/Results: It is difficult to understand, what exactly the OR of 1.15 means in terms of survival. I suggest adding a sentence explaining this in simple words. The addition of separate results for hazard ratio-studies increases the confusion. Please make both numbers more accessible by an explaining sentence.

Table 1/2: 

Please report descriptive data about time of survival or the time period studied for intervention and control condition for OR and HR studies. 

The tables should also entail brief information about the content and type of the intervention in addition to the format.

The tables would also be easier to read if the confidence interval were included rather than or in addition to the standard error.

More information is needed on how the authors decided on whether a study was "effective" or "ineffective" regarding psychosocial outcomes. Most studies may have been ineffective with regard to some and effective with regard to other outcomes. 

Figures: Does effectiveness refer to psychosocial outcomes? Please include a note in the figure caption.

What size was the association between effectiveness in terms of psychosocial improvements and risk of bias? Did the authors control for multicollinearity of the two variables entered simultaneously to the regression models? 

3) A highly relevant result is that if high-quality studies were considered alone, no effect on survival was found. This is an important finding suggesting cautious interpretation. 

The conclusion to better meet patients' needs does not follow from the results.

Reviewer #3: Review, Psychosocial support Interventions and medical survival 

This is a very comprehensive, large, and rigorous meta analysis of psychosocial support interventions and survival for medical patients (mostly CVD and cancer). A total of over 40,000 studies were screened, and 140 studies were included in the meta-analysis. The approach adopted for study selection and coding as well as the careful analytic approach taken are key strengths of this meta analysis. The expansion to multiple diseases is novel, and the inclusion of intervention attributes is also novel and important. The conclusions are meaningful and likely to have an impact on clinical care.

I have a few comments, which are mostly minor.

1) In the first sentence, the consensus report calling for interventions for loneliness and social isolation in medical and health care settings is interesting, but the construct of social isolation and loneliness among persons dealing with chronic illnesses. While a small distinction, if the statement is true, then it might be helpful to change the reference to something more relevant.

2) Some interventions contain content that focuses on enhancing support within the person's network using CBT (overcoming barriers to support). Other interventions are delivered by aj interventionist but would contain solely education and skill practice, but like most 1:1 treatments, a bond develops. Some group treatments are not support oriented, in that the group is more of an education delivery method, but the members obtain support after group or during group exercises in the room. Solely online interventions can contain supportive elements, via email exchanges/message boards, or even videotaped content offering normalization and narratives from other patients. Disease management support is categorized as support, but these are more educational interventions, unless I am missing something. Couples treatments may or may not include support, but may reduce conflict. It would have been very helpful to define social, emotional, and psychological support more clearly and discuss why disease management support is included. Disease management is assumedly to foster adherence to medical treatment. The supplemental materials define interventions but the above approaches would not be clearly included or excluded as the definition is very general.-

3) The conclusion that the interventions that resulted in psychological improvement achieved better outcomes in terms of survival than control interventions seems a little generic. Since the interventions may have provided support as well as coping skills (at least if my understanding of the interventions meeting criteria are correct), then the conclusion might state something more along the lines of "the intervention may not have included effective components." This leads me to the broader concern that the psychotherapeutic goals of each intervention might be better characterized in this study, as efficacy may be due to the components, and that would guide this study's implications much more clearly. 

4) It might have been interesting to discuss the baseline level of loneliness/low support as a moderator of effects for future studies that focus on survival. It may be that the interventions have a stronger impact among lonelier patients, and that may be why older patients fared better.

[LINK]

---

## [Decision Letter · Decision Letter 2]

8 Mar 2021

Dear Dr. Smith,

Thank you very much for re-submitting your manuscript "Psychosocial Support Interventions and Medical Patient Survival: A Meta-Analysis of 106 Randomised Controlled Trials" (PMEDICINE-D-20-03297R2) for review by PLOS Medicine.

I have discussed the paper with my colleagues and the academic editor and it was also seen again by one of the original reviewers. I am pleased to say that provided the remaining editorial and production issues are dealt with we are planning to accept the paper for publication in the journal.

[LINK]

We look forward to receiving the revised manuscript by Mar 15 2021 11:59PM.   

Sincerely,

Caitlin Moyer, Ph.D.

Associate Editor 

PLOS Medicine

plosmedicine.org

Requests from Editors:

1. Please revise the title to indicate that the meta-analysis was restricted to studies where recruitment occurred in health care settings, e.g. "Psychosocial support interventions delivered in inpatient and outpatient health care settings: a meta-analysis of 106 randomized controlled trials"

Please ensure that the Abstract similarly emphasizes this distinction.

2.Data availability statement: Thank you for making the study data available. Please provide a complete link to the dataset here (for example: https://osf.io/3qydb/)

3.Abstract: Conclusions: Line 57: The link to the dataset registration can be removed from the abstract.

4.Throughout: Please remove spaces from within brackets of in-text citations ([1,2] instead of [1, 2] for example)

5.Introduction: Line 140: Please revise to “sought to evaluate”

6.Methods: Line 152: Please include a reference to the supporting information file containing the PRISMA checklist (S1_Checklist)

Methods: Line 170/Table 1: Please include a very brief rationale for excluding palliative care interventions, in the most appropriate point in the text (perhaps drawing on the most relevant points from the updated cover letter).

7.Methods: Data analysis section: The use of the first person here is fine; however, if possible we suggest considering a reduction in the use of “We…” as this seems to start the beginning of most sentences.

8.Table 1: Please consider if this information needs to be displayed in a table format, or can be presented as subsections within the text of the Methods section.

9.Figure 1: Please provide a more descriptive title/legend for the flow diagram.

10. Figure 2: Please provide a more descriptive title/legend for the diagram illustrating risk of bias.

11. Results: Description of included studies: Please provide the numerators/denominators (or at least the number, as the total number of studies, 106, is given earlier in the paragraph) for any percentages reported where the numbers aren’t displayed in a table.

12. Results: Main analyses: Please report p<0.001 (rather than p=0.0001 at line 248, for example) unless there is a reason to report them to this number of digits.

13. Results: Line 344-345 and Line 348-349: Please provide the p value in addition to the 95% CIs for the publication bias analysis for the 4 and 8 imputed studies from the HR and OR data.

14. Page 24: Please remove the Competing Interests and Funding sections from the main text, and ensure that all information is accurately entered in the “Competing Interests” and “Financial Disclosures” section of the manuscript submission system.

15. Reference list: Please use the "Vancouver" style for reference formatting, and see our website for other reference guidelines https://journals.plos.org/plosmedicine/s/submission-guidelines#loc-references

16. S1 Appendix: It would be helpful to provide the search criteria, analysis plan, and PRISMA checklist as separate files.

17. S2 Appendix: Please provide legends in addition to titles for all figures and tables, including those in Supporting Information files. Again, it would be helpful for each table/figure to be a separate file, to make it easier to refer the reader to the correct figure within the text.

Comments from Reviewers:

Reviewer #1: The authors have addressed my concerns and I now recommend publication

Peter Flom

[LINK]

---

## [Editor Report · Decision Letter 3]

25 Mar 2021

Dear Dr Smith, 

On behalf of my colleagues and the Academic Editor, Alexander C. Tsai, I am pleased to inform you that we have agreed to publish your manuscript "Effects of Psychosocial Support Interventions on Survival in Inpatient and Outpatient Health Care Settings: A Meta-Analysis of 106 Randomised Controlled Trials" (PMEDICINE-D-20-03297R3) in PLOS Medicine.

PRESS

Sincerely, 

Caitlin Moyer, Ph.D. 

Associate Editor 

PLOS Medicine